# Transient photocurrents in a subthreshold evidence accumulator accelerate perceptual decisions

Timothy L. H. Wong [1,2], Clifford B. Talbot [1,2] & Gero Miesenböck [1] ✉

Perceptual decisions are complete when a continuously updated score of sensory evidence reaches a threshold. In *Drosophila*, αβ core Kenyon cells (αβ$_c$ KCs) of the mushroom bodies integrate odor-evoked synaptic inputs to spike threshold at rates that parallel the speed of olfactory choices. Here we perform a causal test of the idea that the biophysical process of synaptic integration underlies the psychophysical process of bounded evidence accumulation in this system. Injections of single brief, EPSP-like depolarizations into the dendrites of αβ$_c$ KCs during odor discrimination, using closed-loop control of a targeted opsin, accelerate decision times at a marginal cost of accuracy. Model comparisons favor a mechanism of temporal integration over extrema detection and suggest that the optogenetically evoked quanta are added to a growing total of sensory evidence, effectively lowering the decision bound. The subthreshold voltage dynamics of αβ$_c$ KCs thus form an accumulator memory for sequential samples of information.

In his essay 'The Problem of Serial Order in Behavior,' Karl Lashley highlighted the ubiquity of neural processes that unfold over time, citing as examples language, navigation, and motor control[1]. Temporal processing is integral also to decision-making because the information necessary to commit to a choice is often collected sequentially[2,3]. Just as a single note will not distinguish two musical scores but a few bars may, the quality of inference improves with the number of observations.

Most studies of how the brain combines multiple pieces of evidence over time have examined the spike rates of individual neurons[3]. In monkeys discriminating directions of visual motion, neurons of different brain areas exhibit firing rate dynamics that correlate with the quality and strength of the sensory stimulus[4,5] or the animal's perceptual judgment[6–8] and its behavioral expression[9]. In the middle temporal visual area (MT or V5), physiologists can decode noisy motion stimuli by counting the spikes of neuron–antineuron pairs attuned to opposite directions of movement[4]—a function perhaps also performed by cells in the lateral intraparietal area (LIP), whose mean firing rates ramp up like time integrals of MT spikes[6–8].

Putative integrators of sensory data supplied by neurons and antineurons are also found in *Drosophila*, but with the important distinction that accumulating evidence appears encoded in graded, subthreshold voltage changes rather than discrete, suprathreshold spikes[10]. During odor intensity discrimination, flies engage ~160 αβ core Kenyon cells (αβ$_c$ KCs)[11], a genetically and anatomically distinct subset of the ~2000 intrinsic neurons that form the mushroom bodies[12,13]. Two opposing and mutually inhibitory populations of on and off αβ$_c$ KCs are thought to integrate sensory-evoked synaptic quanta to spike threshold and thereby collect evidence for increases or decreases in odor concentration[10,14]. The ratio of trials during which one or the other population spikes first correlates with decision accuracy, and the average latency of the first spike correlates with decision speed[10]. Despite the success of these neurometric predictions, however, it remains unclear whether the activity of αβ$_c$ KCs dictates the organism's decisions or merely echoes them. Here, we test the causal influence and precise role of αβ$_c$ KCs by manipulating their membrane potentials with closed-loop optogenetics during odor discrimination.

[1]Centre for Neural Circuits and Behavior, University of Oxford, Tinsley Building, Mansfield Road, Oxford OX1 3SR, UK. [2]These authors contributed equally: Timothy L. H. Wong, Clifford B. Talbot. ✉e-mail: gero.miesenboeck@cncb.ox.ac.uk

## Results and discussion

### Generation of EPSP-like photocurrents

Optical microstimulation of either the on or the off KC population should result in coupled changes in reaction time and choice accuracy if the difference in the membrane potentials of these neurons represents the decision variable. The two opponent KC pools, however, elude clean optogenetic control because their membership is neither fixed nor genetically determined; the functional partition of the KC ensemble arises from differential connectivity rather than differential gene expression and varies from odor to odor and individual to individual[13–15]. Nonetheless, since any increase in odor concentration will raise the membrane potentials of on cells closer to spiking than that of off cells[10], the simultaneous addition of small, EPSC-like currents should, in principle, nudge on cells, but not off cells, across action potential threshold. Because voltage changes must pass through the nonlinearity of spike generation to have behavioral impact[10,14], subthreshold photostimulation of the entire $\alpha\beta_c$ KC population could thus provide a selective advantage to the on KC pool during increases in odor concentration, and vice versa, without directly firing the neurons to force an immediate choice.

EPSC-like photocurrents were generated by targeting the red-shifted optogenetic actuator[16,17] CsChrimson to the dendrites of $\alpha\beta_c$ KCs. We borrowed potential localization motifs from voltage-gated potassium channels[18,19], cell adhesion molecules[19–23], or neurotransmitter receptors[24,25] (Fig. 1a) and appended them to CsChrimson::tdTomato via a flexible C-terminal linker (Fig. 1b). Of the eight motifs tested, only the one transplanted from the AMPA receptor subunit GluR1[24] redirected CsChrimson from what appeared to be a predominantly axonal location in the mushroom body lobes (but whose intense fluorescence may in reality reflect the high packing density of KC axons) to the somatodendritic compartment, the calyces of $\alpha\beta_c$ KCs (Fig. 1c, d, Supplementary Fig. 1). The CsChrimson fusion protein carrying the ineffective motif[21] derived from neuroligin 1 (NLGN1) served as our untargeted control in all subsequent electrophysiological and behavioral experiments.

The success of the GluR1 signal in localizing opsin expression, as well as the failures of the other seven sequences we tested, defy simple explanations. At the outset, we would have bet on information contained in the somatodendritically located potassium channel Shal, which is native to $\alpha\beta_c$ KCs[10,26], as most directly compatible with the protein sorting machinery of these neurons, but alas, the particular sequence we chose went unrecognized. The effective GluR1 signal, in contrast, lacks the well-defined tyrosine/di-leucine localization motif of Shal[18] or a PSD-95/Dlg/ZO-1 domain[24] and failed to target channelrhodopsin-2 to the dendritic fields of mouse retinal ganglion cells where, ironically, the in our hands unsuccessful NLGN1 motif prevailed[27].

$\alpha\beta_c$ KCs expressing dendritically targeted or untargeted CsChrimson responded to 25-ms pulses of red light with depolarizations (Fig. 1e) whose amplitudes varied with light intensity and the subcellular location of the channel (Fig. 1f). At the optical powers used in later behavioral analyses (0.01–0.20 mW mm$^{-2}$), CsChrimson::GluR1 produced only small, subthreshold membrane potential changes of 1–4 mV at the soma of $\alpha\beta_c$ KCs (Fig. 1f), equivalent to at most three odor-evoked synaptic quanta[10,13].

### Optical microstimulation of $\alpha\beta_c$ KCs shortens reaction times

Flies were trained to avoid a high concentration (10 ppm) of 4-methylcyclohexanol (MCH) and had to discriminate the reinforced concentration from a lower intensity of the same odor[10,11]. The difficulty of discrimination was titrated by varying the MCH concentration ratio during testing. Figure 2 plots mean reaction times—that is, averages of the time spent in the central decision zone during each trial[11]—and decision accuracies, which are expressed as the probability of choosing left, as functions of the odor concentration ratio on a

logarithmic scale; the extremes of −1 and +1 represent the lowest difficulty levels at respective odor contrasts of 1:10 and 10:1. The chronometric and psychometric functions displayed their characteristic dome and sigmoidal shapes, respectively, in the absence of stimulating light (Fig. 2a, b). Mean reaction times increased with diminishing stimulus contrast by approximately twofold from the easiest to the hardest difficulty levels (Fig. 2a); accuracy peaked at ~97% in the easiest discriminations and dropped to near chance in the most difficult tasks (Fig. 2b).

In photostimulation trials, entry into the decision zone triggered a single 25-ms pulse of red light at low (0.01 mW mm$^{-2}$) or high (0.20 mW mm$^{-2}$) intensity (Supplementary Fig. 2). Compared to control trials, the injection of depolarizing current into the $\alpha\beta_c$ KC dendrites of flies expressing GluR1-targeted CsChrimson under *NP6024-GAL4* control[11,12] lowered the central peak but not the flanks of the chronometric function in a photon dose-dependent fashion (Fig. 2a); in other words, reaction times were shortened most in the most difficult tasks. This difficulty-dependent acceleration of the decision process suggests that strong, unambiguous sensory evidence drove the membrane potentials of $\alpha\beta_c$ KCs to threshold before the extra light-induced depolarization could have a perceptible effect, whereas the impact of photostimulation was more salient in difficult discriminations, where a wider voltage gap remained to be closed. The decrease in mean reaction time was linked to what appeared to be a subtle decline in choice accuracy at intermediate and high difficulties (i.e., 7 or 9 vs. 10 ppm; Fig. 2b), as would be expected if the optogenetic intervention curtailed the evidence collection period.

Given that small dendritic photocurrents could speed up challenging decisions, albeit at a cost of marginally compromised accuracy, we asked if membrane potential deflections large enough to push $\alpha\beta_c$ KCs near or across spike threshold would precipitate instant but random choices, effectively flattening both the chronometric and psychometric curves. In flies expressing CsChrimson::NLGN1, red light indeed caused a drastic decrease in reaction times at high intensities (Fig. 2c) and compressed the psychometric function to near chance level in otherwise only moderately difficult discriminations; even the maximal accuracy dropped from 97 to 87% (Fig. 2d). Nonetheless, a measure of difficulty dependence persisted, indicating that behavioral choice could not be entirely decoupled from sensory input. Of course, a significant unknown—and unknowable—in these experiments is the trial-to-trial reliability and speed with which untargeted CsChrimson elicited $\alpha\beta_c$ KC spikes in the intact, behaving fly.

To exclude a visual response to the optical stimulus, we tested flies expressing CsChrimson::GluR1 in $\alpha'\beta'$ KCs, using the *VT030604-GAL4* driver[28], as well as flies carrying the core KC driver[11,12] *NP6024-GAL4* but lacking a *UAS−CsChrimson* responder transgene. $\alpha'\beta'$ KCs are naturally poor temporal integrators[10] and, judging from manipulations of their biophysical properties, are uninvolved in the evidence accumulation step of odor discrimination[10]. Despite greater scatter in the reaction time data of $\alpha'\beta'$ KC > CsChrimson flies, both in the absence and presence of light, the overall form of their chronometric functions was unaltered by photostimulation (Fig. 3a); and despite shallower psychometric curves and lower maximal accuracy relative to the $\alpha\beta_c$ KC experimental groups (again pointing to a genotype effect linked to the *VT030604-GAL4* strain), illumination failed to change the mean choice accuracies at all difficulty levels (Fig. 3b). In the no-CsChrimson control, the chronometric and psychometric functions similarly overlapped in the unstimulated and photostimulated conditions (Fig. 3c, d).

### Temporal integration vs. extrema detection

Two sequential sampling models sought to formalize the role of $\alpha\beta_c$ KCs in the decision process: a drift-diffusion model, which assumes integration of noisy sensory evidence over time to a decision criterion[10,11,29–31], and an extrema detection model, where successive samples of evidence are drawn from a stationary distribution until one

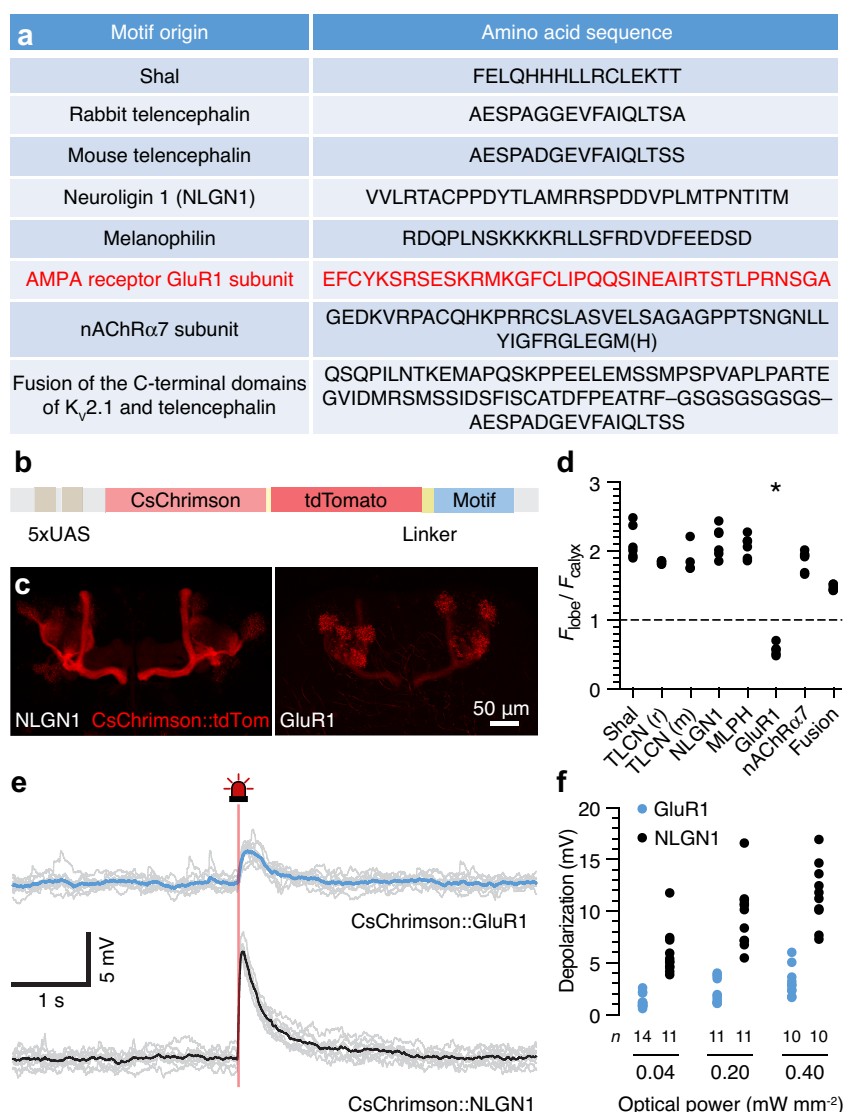

**Fig. 1 | Dendritically targeted CsChrimson produces EPSP-like depolarizations.**
**a** Origins and amino acid sequences of eight putative dendritic targeting motifs. The effective localization motif derived from the AMPA receptor subunit GluR1 is shown in red type. **b** Structure of the *UAS* transgenes encoding fusion proteins of CsChrimson, tdTomato, and C-terminal targeting motifs. **c** Native tdTomato fluorescence in the axonal αβ lobes (left) or dendritic calyces (right) of the mushroom bodies in flies expressing *NP6024-GAL4*-driven *UAS* transgenes encoding CsChrimson::NLGN1 (left) or CsChrimson::GluR1 (right). The images show representative examples of 6 and 5 brains, respectively. **d** Fluorescence intensity ratios of mushroom body lobes and calyces in flies expressing CsChrimson::motif fusion proteins. Data points are averages of both hemispheres in a brain. The asterisk indicates that the localization of CsChrimson::GluR1 differed from that of all other constructs ($F_{7,36} = 53.767$, $P < 0.0001$, one-way ANOVA; $q_{36} = 15.6951$, 12.2384,

12.0416, 15.7890, 15.0769, 13.0373, and 9.0922 for pairwise post-hoc comparisons of the GluR1 motif with sequences derived from Shal, TLCN (r), TLCN (m), NLGN1, MLPH, nAChRα7, and the fusion of TLCN and $K_v2.1$, respectively, by two-sided Dunnett's test, multiplicity-adjusted $P < 0.0001$ throughout). **e** Example membrane potential traces (bold lines indicate averages of 10 individual sweeps, shown in gray) of αβc KCs expressing CsChrimson::GluR1 (top) or CsChrimson::NLGN1 (bottom) during 25-ms pulses of red light (0.20 mW mm⁻²). **f** Maximal depolarizations evoked by 25-ms pulses of red light at 0.04, 0.20, and 0.40 mW mm⁻² in αβc KCs expressing CsChrimson::GluR1 (blue) or CsChrimson::NLGN1 (black). Data points are averages of 10 sweeps in single αβc KCs. Two-way ANOVA detected effects of CsChrimson's subcellular location ($F_{1,61} = 177.511$, $P < 0.0001$) and illumination intensity ($F_{2,61} = 19.215$, $P < 0.0001$) as well as a localization × intensity interaction ($F_{2,61} = 4.815$, $P = 0.0114$). Source data are provided as a Source Data file.

exceeds the criterion[31,32] (Fig. 4a). Both models were specified in terms of four free parameters and fit to the full reaction time and choice accuracy distributions, using Bayesian estimation. The free parameters were *A*, the height of the decision criterion; *k*, a scaling factor that linearly relates task difficulty *x* to drift rate *v* (in the case of drift-diffusion) or the mean μ of the evidence distribution (in the case of extrema detection); $T_0$, the non-decision or residual time[10,11,29–31]; and its standard deviation $\sigma_0$.

To identify the most likely decision strategy and the impact of optogenetic KC microstimulation on it, we computed differences in the Bayesian information criterion (ΔBIC) between pairs of models, a quantity connected to the probability ratio of the observed data,

conditioned on one or the other model (the Bayes factor)[33,34]. Initial comparisons of drift-diffusion and extrema detection fits to reaction time and choice accuracy distributions in the unstimulated condition decisively[34] favored an integration strategy for all genotypes (Fig. 4b); subsequent attempts to pinpoint the effect of photostimulation therefore focused exclusively on different types of drift-diffusion model. All models kept the non-decision parameters $T_0$ and $\sigma_0$ constant but allowed either bound height *A*, scaling factor *k*, both, or neither to vary with optical power (Fig. 4c). Under photostimulation conditions, a model with light-sensitive *A* and/or *k* captured the choice behavior of flies expressing CsChrimson in αβc KCs better than a light-insensitive model did, whereas a model with no light-sensitive

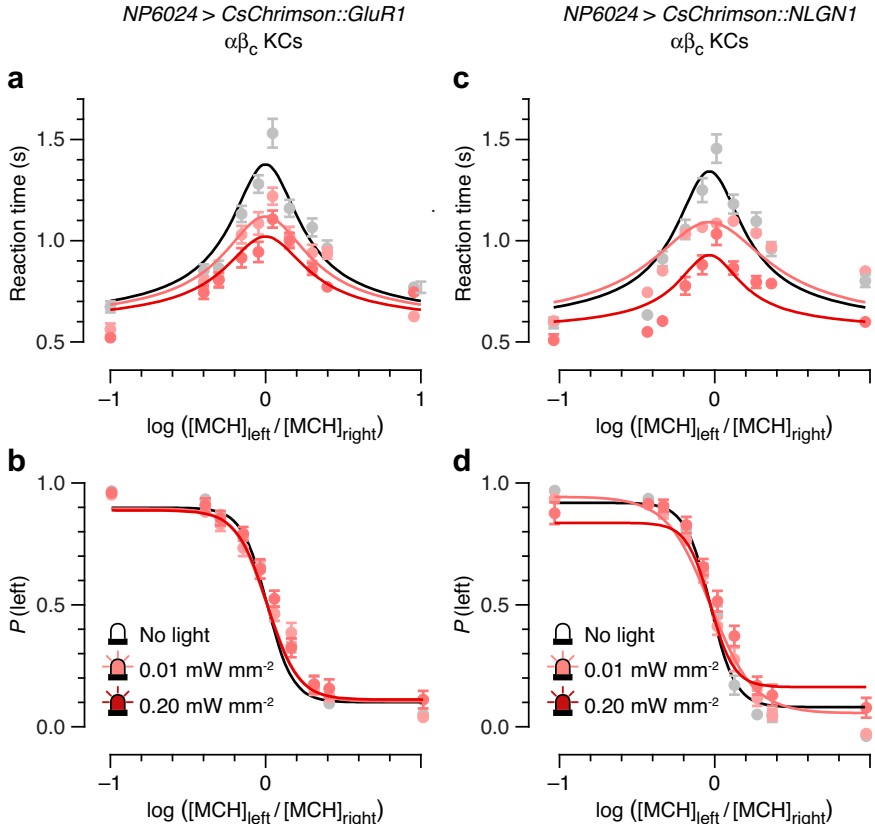

**Fig. 2 | Optogenetic microstimulation of $\alpha\beta_c$ KCs shortens reaction times.** Black and gray, unstimulated conditions; pink, low-intensity photostimulation (25 ms, 0.01 mW mm$^{-2}$); red, high-intensity photostimulation (25 ms, 0.20 mW mm$^{-2}$). **a**, **b** Reaction times (**a**) and choice accuracies (**b**) as functions of MCH concentration ratio in flies expressing *NP6024-GAL4*-driven CsChrimson::GluR1 in $\alpha\beta_c$ KCs. **c**, **d** Reaction times (**c**) and choice accuracies (**d**) as functions of MCH concentration ratio in flies expressing *NP6024-GAL4*-driven CsChrimson::NLGN1 in $\alpha\beta_c$ KCs. Data

are means ± SEM of the reaction times and choice accuracies observed in *n* decisions by *m* flies (see Supplementary Table 1 for sample sizes). Solid lines indicate simultaneous least-squares fits of a drift-diffusion model to reaction time and accuracy data. Bayesian inference offers decisive support for a photostimulation effect on reaction time and choice accuracy (Bayes factor <0.0001; see Fig. 4d). Source data are provided as a Source Data file.

parameters was preferred (by a smaller ΔBIC, but still a decisive Bayes factor on Jeffreys' scale[34]) for flies expressing CsChrimson in α'β' KCs and no-CsChrimson controls (Fig. 4d). To influence the decision process, optical stimuli thus needed to perturb the membrane potentials of $\alpha\beta_c$ KCs.

To determine whether a change in bound height *A* or scaling factor *k* was responsible for the observed shortening of reaction times, we constructed models whose light-sensitivity derived exclusively from variation in one or the other parameter and found ΔBIC overwhelmingly in favor of variable bound height (Fig. 4e). Plots of *A* showed narrow posterior distributions with sharp, well-defined maxima that declined with increasing optical power—progressively in the case of CsChrimson::GluR1, and precipitously at high illumination intensity in the case of CsChrimson::NLGN1 (Fig. 4f). This light-induced decrease in bound height is conceptually identical to an offset in the decision variable, which represents the accumulated evidence, and therefore makes $\alpha\beta_c$ KCs the likely site of temporal integration. The same line of reasoning—a distinction between stimulation-induced changes in bound height and scaling factor—has been advanced to support the notion that lateral interparietal (LIP) cortical neurons integrate motion direction signals supplied by middle temporal (MT) neurons[35]: electrical microstimulation of LIP lowered the decision bound (that is, it brought the decision variable closer to criterion), whereas stimulation of MT increased the drift rate of the diffusion process (that is, it augmented the stream of momentary evidence). A critical difference between these experiments and ours, however, is that cortical neurons were stimulated for the entire period from the

first appearance of the visual display to the saccade indicating the monkey's choice[35]. We, in contrast, inserted a single 25-ms depolarization into $\alpha\beta_c$ KCs and found that its impact persisted until the fly made its decision, consistent with a process of temporal integration.

## The case for subthreshold integration

Targeting CsChrimson to the dendrites of $\alpha\beta_c$ KCs allowed a pulse of light to lift the somatic membrane potential by at most a few millivolts (Fig. 1e, f), far short of the ~10-mV depolarization needed to breach spike threshold[10,13]. Because opaque tissue in the optical path was removed for electrode access in these recordings, the light-induced membrane potential changes in the intact fly were almost certainly even smaller—but still able to influence the speed of odor discrimination (Fig. 2) by biasing the decision variable (Fig. 4). Our experiments thus strengthen the increasingly compelling case that $\alpha\beta_c$ KCs gather and retain sensory evidence by integrating subthreshold synaptic potentials rather than spikes. This case initially rested on a correlation between the average latencies of the first odor-evoked $\alpha\beta_c$ KC spikes and mean reaction times[10] and received strong indirect support from the discovery that $\alpha\beta_c$ KCs communicate in two distinct modes with different synaptic partners at different stages of the decision process[14]. During integration, when synaptic input from olfactory projection neurons drives the membrane potentials of on and off KCs toward spike threshold, the two neuronal populations compete with each other by recruiting pooled inhibitory feedback from a GABAergic interneuron via graded-potential synapses, which transmit in the absence of action potential discharges[14]. The result of the competition,

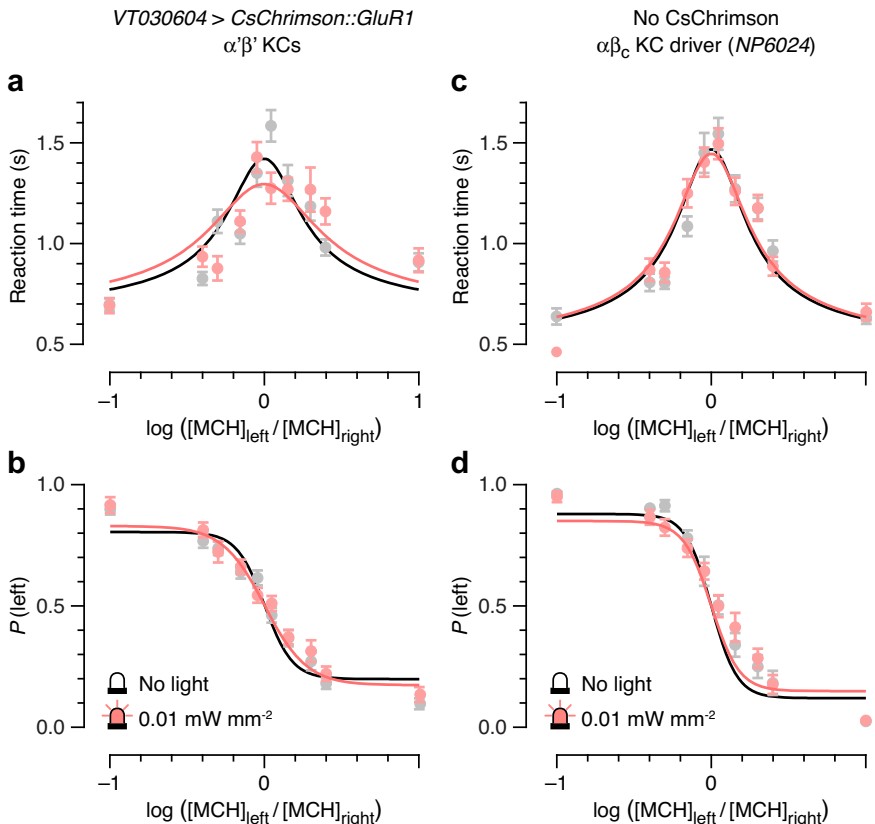

**Fig. 3 | Optogenetic microstimulation of α'β' KCs, or light alone, does not shorten reaction times.** Black and gray, unstimulated conditions; pink, low-intensity photostimulation (25 ms, 0.01 mW mm⁻²). **a, b** Reaction times (**a**) and choice accuracies (**b**) as functions of MCH concentration ratio in flies expressing *VT030604-GAL4*-driven CsChrimson::GluR1 in α'β' KCs. **c, d** Reaction times (**c**) and choice accuracies (**d**) as functions of MCH concentration ratio in flies carrying the *NP6024-GAL4* driver but lacking the *UAS-CsChrimson::GluR1* transgene. Data are

means ± SEM of the reaction times and choice accuracies observed in *n* decisions by *m* flies (see Supplementary Table 2 for sample sizes). Solid lines indicate simultaneous least-squares fits of a drift-diffusion model to reaction time and accuracy data. Bayesian inference offers decisive support against a photostimulation effect on reaction time and choice accuracy (Bayes factor > 1000; see Fig. 4d). Source data are provided as a Source Data file.

however, is conveyed to mushroom body output neurons solely by KC spikes[14]. Subthreshold voltages are thus the principal information carriers while the decision forms, and an action potential marks its completion.

## Methods

### Fly strains
Fly strains were cultivated on standard cornmeal agar under a 12 h light:12 h dark cycle at 20 °C. The driver lines *NP6024-GAL4* and *VT030604-GAL4* were used to target transgene expression to αβ_c and α'β' KCs, respectively[12,28]. *UAS–CsChrimson* transgenes contained one of eight dendritic targeting motifs that had been codon-optimized for *Drosophila*. Shorter motifs derived from Shal and telencephalins[18,20,22] (Fig. 1a) were attached to the CsChrimson::tdTomato coding sequence[17] by PCR, using primers that incorporated the motif plus a flexible linker (SGGSGG). All remaining, longer motifs[19,21,23–25] (Fig. 1a) were synthesized de novo (GeneArt, ThermoFisher) and appended to CsChrimson::tdTomato by fusion PCR (Supplementary Table 3). The final amplified products were ligated to *pJFRC-MUH*[36] and integrated into landing sites[37] *attP40* or *attP2* (BestGene). Experimental flies were heterozygous for all transgenes, raised in darkness, and transferred to food supplemented with 0.5–0.6 mM all-*trans* retinal (Sigma-Aldrich) in dimethyl sulfoxide 18–24 h before training and testing.

### Fluorescence imaging
Brains of adult flies were dissected in phosphate-buffered saline (PBS; 137 mM NaCl, 3 mM KCl, 8 mM Na₂HPO4, 1.5 mM KH₂PO4, pH 7.3),

fixed in 4% (w/v) paraformaldehyde in PBS for 15 min at room temperature, washed three times in PBS for 15 minutes, mounted in Vectashield (Vector Laboratories), and imaged on a Leica TCS SP5 confocal microscope with an HCX IRAPO L 25×/0.95 W objective (Leica). Native tdTomato fluorescence was quantified in maximum intensity projections of confocal image stacks containing the αβ lobes or calyces. The average grayscale in three manually defined regions of interest per hemisphere was measured in ImageJ. These six values were averaged, and the mean of three background fluorescence estimates was subtracted to give $F_{lobe}$ or $F_{calyx}$.

### Electrophysiology
For whole-cell patch-clamp recordings in vivo[10], male flies aged 1–2 days were fixed to a custom mount with soft thermoplastic wax (Agar Scientific). Cuticle, trachea, adipose tissue, and perineural sheath were surgically removed in a window large enough to provide access to αβ_c KCs in the dorsal brain. Extracellular solution (pH 7.3) containing 5 mM TES, 103 mM NaCl, 3 mM KCl, 26 mM NaHCO₃, 1 mM NaH₂PO4, 1.5 mM CaCl₂, 4 mM MgCl₂, 8 mM trehalose, 10 mM glucose, and 7 mM sucrose (275 mOsM, equilibrated with 5% CO₂ and 95% O₂) continuously superfused the brain. Patch pipettes (14–18 MΩ) were fabricated from borosilicate glass capillaries with outer and inner diameters of 1.5 and 0.86 mm (Sutter Instruments), using a DMZ universal electrode puller (Zeitz), and filled with solution (pH 7.3) containing 10 mM HEPES, 140 mM potassium aspartate, 1 mM KCl, 4 mM MgATP, 0.5 mM Na₃GTP, and 1 mM EGTA (265 mOsM). Pipettes were visually targeted to red-fluorescent KC somata using a combination of

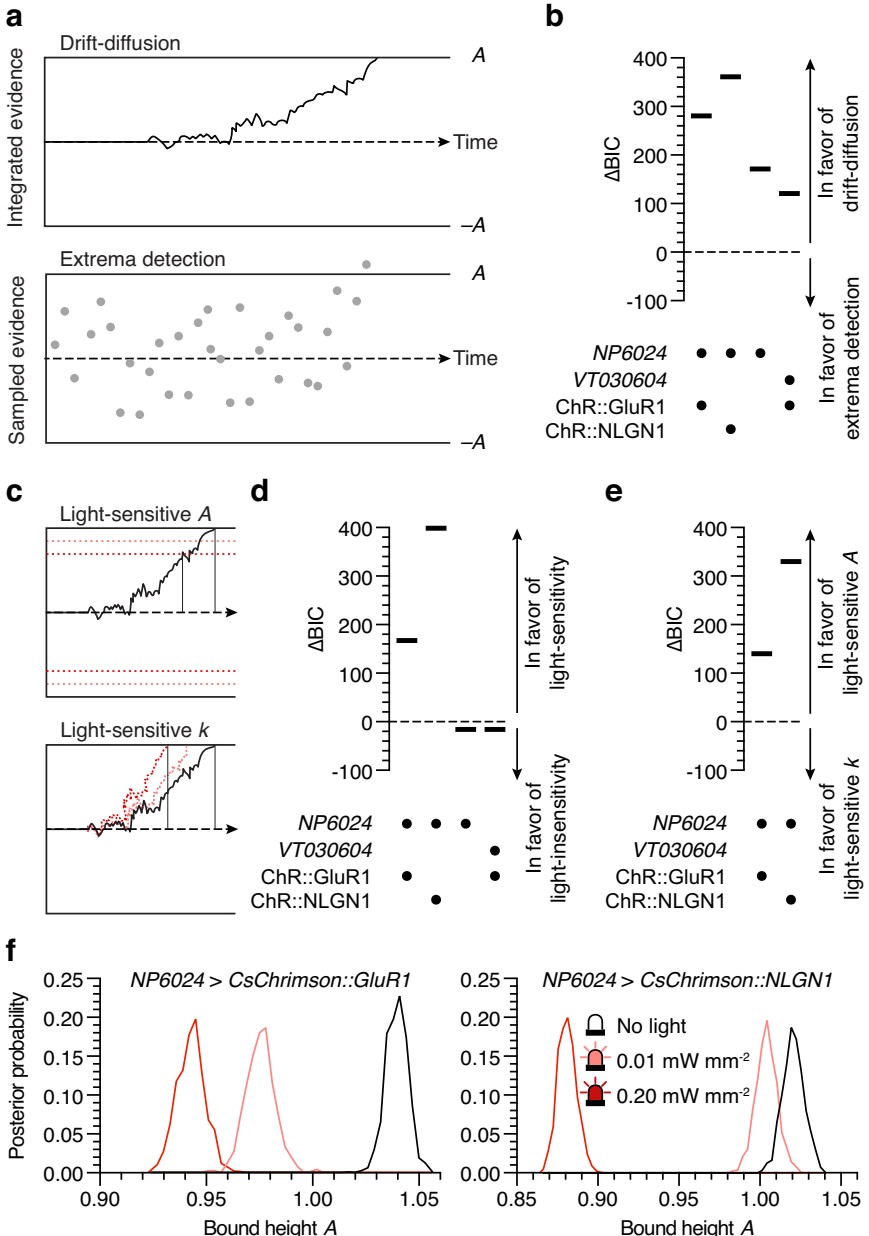

**Fig. 4 | Optogenetic microstimulation of αβ_c KCs lowers the decision bound.**
**a** Sequential sampling models: drift-diffusion (top) and extrema detection (bottom). **b** Comparison of drift-diffusion and extrema detection models in the indicated genotypes, in the absence of stimulation light. **c** Models of how optogenetic microstimulation of αβ_c KCs shortens reaction times: changes in bound height $A$ (top) or scaling factor $k$ (bottom). **d** Comparison of light-sensitive and light-insensitive drift-diffusion models for the indicated genotypes, in the presence of stimulation light. **e** Comparison of drift-diffusion models with light-sensitive bound height $A$ or scaling factor $k$ for the indicated genotypes, in the presence of stimulation light. **f** Posterior probability distributions of bound height $A$ in flies expressing *NP6024-GAL4*-driven CsChrimson::GluR1 (left) or CsChrimson::NLGN1 (right) in αβ_c KCs. Black, unstimulated conditions; pink, low-intensity photostimulation (25 ms, 0.01 mW mm⁻²); red, high-intensity photostimulation (25 ms, 0.20 mW mm⁻²). Source data are provided as a Source Data file.

epifluorescence and differential interference contrast on an Axioscop 2 FS mot microscope (Zeiss) equipped with a 40×/0.8 W objective (44 00 90, Zeiss) and an X-Cite 120PC Q light source (Excelitas Technologies). A red LED (625 nm, M625L3, Thorlabs) was mounted on an aspheric condenser lens ($f = 20.1$ mm, ACP2520-A, Thorlabs) ~10 cm from the head of the fly and powered by a Thorlabs LEDD1B driver.

Signals were acquired at room temperature (21–23 °C) with a MultiClamp 700B amplifier (Molecular Devices), lowpass-filtered at 10 kHz, and sampled at 50 kHz using a Digidata 1440 A digitizer controlled through pCLAMP 10 (Molecular Devices). Data were corrected for liquid junction potential and analyzed with custom procedures using the NeuroMatic package in Igor Pro 8 (WaveMetrics). Bridge

balance was used to compensate for series resistance. The most negative membrane potential recorded immediately after break-in, in the absence of a holding current and without correcting for errors introduced through seal conductances, was taken to represent the resting potential. For optogenetic stimulation, 25-ms pulses of red light were delivered in sweeps of ten.

**Measurement of reaction times and decision accuracies**
Male flies aged 7–10 days were trained individually in plexiglass chambers (50 mm long, 5 mm wide, 1.3 mm high) with printed circuit boards (PCBs) as floors and ceilings[11,38]. Twelve consecutive 15-s odor presentations of 10 ppm 4-methylcyclohexanol (MCH, Sigma-Aldrich),

interspersed with 15-s presentations of clean air, were paired with electric shocks by connecting the PCBs for 1 s to a 70-V source. The trained flies were transferred to identical chambers containing float glass strips (UQG Optics) in lieu of PCBs, in a custom-built 6-chamber testing apparatus. Filtered, flow-controlled (CMOSens, Sensirion), and humidified carrier air was mixed with flow-controlled MCH streams to yield the desired odor concentrations at final flow rates of 0.25 l min$^{-1}$ per half-chamber. The same odor delivery system was used during training and testing. The delivery of odors and electric shock was controlled by a virtual instrument written in LabVIEW 2012 (National Instruments).

During the 2-min testing period (Supplementary Fig. 2), flies had to distinguish MCH at the reinforced concentration of 10 ppm, which perfused one arm of the chamber, from a lower MCH concentration in the other arm (1, 4, 5, 7, or 9 ppm). The odor contrast determined the difficulty of the perceptual decision. The chambers were backlit by 830 nm LEDs (TSHG8400, Vishay) with paper diffusers and individually imaged at 30 frames s$^{-1}$ by dedicated cameras (Raspberry Pi NoIR Camera v2) with 8-mm lenses (MVL8M23, Thorlabs) and IR filters (FL830-10, Thorlabs). Each camera was connected to a headless Raspberry Pi 4 (Model B, 4GB) communicating with a host PC via TCP/IP over a local network. Motion detection software running on each individual Pi extracted real-time position data every three frames by subtracting the most recent video image from a static background acquired before testing. Non-zero pixel clusters in the difference image were intensity- and size-thresholded; the centroid of the largest cluster was taken as the position of the fly. Raspberry Pis were running OS Raspbian 10 and custom software written in Python 3.6.1, with modules installed using Berryconda: scipy 1.0.0, pyserial 3.4, pillow 3.4.1, and picamera 1.13.

For optogenetic stimulation (Supplementary Fig. 2), entry of the fly into an empirically defined, 7 mm wide central decision zone[11] triggered a red LED (624 nm, ILH-GD01-RED1- SC201, Intelligent LED Solutions) emitting a single 25-ms pulse of light at either low (0.01 mW mm$^{-2}$) or high (0.20 mW mm$^{-2}$) intensity via a microcontroller running at 16 MHz (Arduino Mega 2560). Arduino code was written in C using the Arduino IDE version 1.8.10. The LEDs were mounted on aspheric condenser lenses ($f = 16$ mm, ACL25416U-A, Thorlabs) and powered by custom circuits based on a constant-current driver (Recom RCD-24-0.70 Vref). Each experimental round consisted of six randomly chosen flies performing the decision task in parallel at the same perceptual difficulty. Control rounds without stimulation light were interleaved with stimulation trials.

Behavioral data were processed offline in Python 3.7.9, after discarding videos showing excessive grooming or animals making <2 end-to-center runs and smoothing position measurements by computing moving averages of 10 frames. The time between entry into and exit from the central decision zone was taken as the reaction time (Supplementary Fig. 2). Correct (exit into the arm containing the lower MCH concentration; Supplementary Fig. 2) and incorrect choices during the 2-minute test period were combined and used to calculate population averages. For a run into the central zone to be counted as a decision, the fly had to reduce its entry velocity by at least 90% and exit the zone within 15 s. At each difficulty level, individual reaction times were normalized for walking speed, as determined by measuring the transit times across identically sized off-center zones on either side of the decision zone[11]. Normalized reaction times (here simply referred to as 'reaction times') and decision accuracies were pooled and plotted as functions of task difficulty to yield chronometric and psychometric functions, respectively[11].

For display purposes, Figs. 2 and 3 include least-squares fits of these functions, obtained with the help of the Levenberg-Marquardt algorithm implemented in the *minimize* function in Python Lmfit[39], to analytic solutions of a simple drift-diffusion model[11,30] for choice probability $P$ and reaction time $T$:

$$P = \frac{1}{1 + \exp(-2Ak(-x))} \tag{1}$$

and

$$T = \frac{A}{kx}\tanh(Akx) + T_0. \tag{2}$$

Here $A$ is the bound height, $k$ is a linear scaling factor linking perceptual difficulty $x$ to drift rate, and $T_0$ is the non-decision or residual time. Choice probability was normalized by the empirically determined maximal and minimal accuracies, giving

$$P = (1 - P_{max}) + (2P_{max} - 1)\frac{1}{1 + \exp(-2Ak(-x))}. \tag{3}$$

Of the four free parameters, $T_0$ was estimated from behavioral data in the absence of stimulation light and fixed, while $A$, $k$, and $P_{max}$ were allowed to vary with light intensity.

## Bayesian parameter estimation and model comparison

Bayesian inference on models specifying different decision strategies and photostimulation-sensitive parameters was used to analyze the full distributions of reaction times and choice probabilities.

**Drift-diffusion model.** Adapting Fürth's analytic solution to the first-passage problem of a diffusion process with drift[40] for bounds at $\pm A$ and a starting point at zero, the probability distributions for correct (+) and incorrect (−) decision times are

$$P_{DDM}(t, \pm | A, k) = \frac{\pi}{4A^2}\exp\left(2\left(\mp kxA - (kx)^2 t\right)\right)\sum_{j=1}^{\infty} j\exp\left(\frac{j^2\pi^2 t}{8A^2}\right)\sin\left(\frac{j\pi}{2}\right). \tag{4}$$

We used the approximation and MATLAB function of Navarro and Fuss[41] to compute this quantity, using an error level of $10^{-29}$ for the approximation of the infinite sum. The non-decision time and its variation were incorporated by numerical convolution with a Gaussian function with mean $T_0$ and standard deviation $\sigma_0$, followed by numerical integration to yield the reaction time distribution.

**Extrema detection model.** The reaction time distribution in the extrema detection model[31] is given by the convolution of an exponential decision time distribution with time constant $\tau_R$ (the mean decision time) and a Gaussian non-decision time distribution centered at $T_0$ (the mean non-decision time), with width $\sigma_0$

$$P_{ED}(t) = \frac{1}{2\tau_R}\exp\left(-\frac{t - T_0}{\tau_R} + \frac{\sigma_0^2}{2\tau_R^2}\right)\mathrm{erfc}\left(-\frac{t - T_0}{\sigma_0\sqrt{2}} + \frac{\sigma_0}{\tau_R\sqrt{2}}\right), \tag{5}$$

where erfc is the complementary error function.

The mean reaction time depends on the probabilities, $p_+$ and $p_-$, of either bound being exceeded in a single sampling interval $\Delta t$

$$\tau_R(p_+, p_-) = \frac{\Delta t}{p_+ + p_-}; \tag{6}$$

these probabilities, in turn, are determined by the bound height $A$, the perceptual difficulty $x$, and the scaling factor $k$

$$p_\pm(A, k) = \frac{1}{2}\mathrm{erfc}\left(\frac{A \mp kx\Delta t}{\sqrt{2\Delta t}}\right). \tag{7}$$

The probability distributions for correct (+) and incorrect (−) decisions at time $t$ are therefore

$$P_{ED}(t, \pm | A, k, T_0, \sigma_0) = P_{ED}(t | A, k, T_0, \sigma_0) \frac{p_\pm(A, k)}{p_+(A, k) + p_-(A, k)}. \quad (8)$$

We used a value of 0.5 ms for the sampling interval $\Delta t$.

**Bayesian estimation and model comparison.** Model parameter distributions were estimated by Markov-chain Monte-Carlo sampling, using MATLAB's *slicesample* function on logarithmic probability densities. We generated 500 samples per fitted parameter, after a burn-in period of 300 samples; initial guesses were set in manual trials on reduced data sets in the absence of photostimulation. We assumed minimal prior knowledge, restricting parameters to positive values and $T_0$ and $\sigma_0$ to the interval (0,3], with equal prior probabilities within this range. Maxima of the smoothed sampled histograms in parameter space (i.e., not the marginal histograms) were taken as best estimates.

Comparisons of decision strategies involved fitting temporal integration (i.e., drift-diffusion) and extrema detection models with four free parameters each ($A$, $k$, $T_0$, and $\sigma_0$) to reaction time and choice accuracy data in the unstimulated condition. Because this comparison decisively favored temporal integration, we considered only four types of drift-diffusion model in our attempt to delineate the effect of optogenetic microstimulation. The non-decision parameters $T_0$ and $\sigma_0$ were always fixed while bound height $A$ and scaling factor $k$ were either allowed to vary, singly or in combination, or kept constant. We implemented global models by combining models for individual light intensities and restricting each fixed parameter to the same value across all conditions. For example, for an experiment with optical power levels 1 and 2 and a hypothetical model with variable $A$ and fixed $k$, the global probability density would be

$$P_{Global}(t_1, \pm_1, t_2, \pm_2 | A_1, A_2, k) = P(t_1, \pm_1 | A_1, k) \times P(t_2, \pm_2 | A_2, k). \quad (9)$$

To compare different models, we computed the Bayesian information criterion (BIC) for each model and then analyzed pairwise differences in BIC ($\Delta$BIC) to determine the better model. The BIC is a measure of the quality of model fitting. It is a good approximation to the logarithm of the integral of the posterior probability density function over all model parameters, multiplied by a factor[33] of −2, and so is proportional to the log-probability that the model is correct given the data:

$$BIC = q \log(n) - 2\hat{L}. \quad (10)$$

Here, $q$ is the number of fitted parameters (global plus [local × photostimulation conditions]), $n$ is the number of decisions in a data set, and $\hat{L}$ is the maximized log-likelihood function of a model. The BIC penalizes overly complex models and rewards well-defined probability distributions with sharp maxima; lower BIC values are therefore preferred. While the absolute value of an individual BIC is difficult to interpret because it depends, among other factors, on the number of observations and their variance, these dependencies cancel when comparing the BIC values of two competing models that are fit to the same data set ($\Delta$BIC), as is the case in all our analyses.

## Statistics

Statistical detail, including test statistics, degrees of freedom, $P$ values (to four significant digits), or Bayes factors, is reported in figure legends. The investigators were neither blind to group allocation, nor were statistical methods used to predetermine sample sizes, which were based on prior studies[10,11].

Behavioral data were analyzed using Bayesian inference on models specifying different decision strategies and light-sensitive parameters, as described above. Converting $\Delta$BIC into Bayes factors (the ratio of probabilities of the observed data, conditioned on one or the other model) according to the approximation following from Schwarz[33]

$$\text{Bayes factor} \approx \exp\left(-\frac{\Delta BIC}{2}\right), \quad (11)$$

allowed us to place the strength of evidence in support of one model over another into the categories established by Jeffreys[34]. A Bayes factor of <0.01 (or >100, depending on the direction of the comparison, i.e., model A vs. model B or model B vs. model A) is considered decisively in favor of one model or the other[34].

Group means in imaging and electrophysiological experiments were compared by one- or two-way ANOVA, followed by pairwise post-hoc analyses using Dunnett's two-sided multiple comparisons test where indicated, in Prism 9 (GraphPad).

## Reporting summary

Further information on research design is available in the Nature Portfolio Reporting Summary linked to this article.

## Data availability

Behavioral data are also available at https://doi.org/10.5281/zenodo.7853204. Source data are provided with this paper.

## Code availability

Custom code used in this study is available at https://doi.org/10.5281/zenodo.7853204.

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

## Acknowledgements

We thank B. Dickson, K. Ito, and the Bloomington, Kyoto, and Vienna stock centers for flies. E. Boyden and G. Rubin kindly provided plasmids. This work was supported by grants (to G.M.) from Wellcome (209235/Z/17/Z, 106988/Z/15/Z, 090309/Z/09/Z) and the Gatsby Charitable Foundation (GAT3237). T.L.H.W. held a graduate scholarship from the Hong Kong Jockey Club.

## Author contributions

T.L.H.W. performed all experiments. C.B.T. developed instrumentation, software, and the Bayesian analysis. G.M. devised the study, supervised the research, and wrote the paper.

## Competing interests

The authors declare no competing interests.
