## [Peer Review File · Nature Communications]

Transient Photocurrents in a Subthreshold Evidence Accumulator Accelerate Perceptual DecisionsREVIEWER COMMENTS

Reviewer #1 (Remarks to the Author):

Wong et al., investigated evidence accumulation in the *Drosophila* mushroom body. They have previously shown that a specific subset of Kenyon Cells is required for odor discrimination and consists of ON/OFF cell types that integrate odor increase or decrease. Now, they set out to causally prove the evidence accumulation mechanism in these KCs. Therefore, they made a new optogenetic tool for activating only the dendritic region of the KCs. They showed before that once the cells were spiking, they seemed to have reached the decision threshold. Manipulation of the membrane potential via the new optogenetic tool however only provides subthreshold activation in the KC soma, thus pushing the cell towards spiking but not initiating spiking itself. Flies were tested for odor intensity discrimination with varying difficulty levels. The more difficult the task, the longer the reaction time and the lower the decision accuracy. Interestingly, upon activation of the KC dendrites during the most difficult tasks, the flies reduce their reaction time but still perform with the same accuracy. Activating all compartments of the KCs however, results in a general reduction in reaction time and a lower accuracy. The authors further investigated how optogenetic stimulation affects evidence accumulation by comparing their data to different computational models. Within a drift-diffusion model, the manipulation seems to be best explained by a lowering of the decision threshold.

This manuscript is written very clearly, and the data is presented in an understandable way. The authors fabricated new and clever tools to elegantly manipulate subthreshold activity in their cell of interest. The new data very nicely shows how the accumulation of subthreshold neural activity can be used as an evidence accumulation system and might be a general mechanism across phyla. Thus, I recommend this manuscript for publication after a very minor revision.

Minor comments:

The figure panel description in Figure 2 seems to be mixed up, please check!

Reviewer #2 (Remarks to the Author):

Overview

This paper sets out to address a very interesting question: how does the process of synaptic integration underly decision making. The model organism is very well suited to this question and the experiments are done carefully and thoughtfully, and then complimented with insightful modelling work. The paper is compact and streamlined but represents a substantial amount of groundwork in genetic tool development that is quite important for the field. Overall, this work could provide an important benchmark and resource for future work in *Drosophila* and other model organisms.

Abstract

General

Abstract is well written and articulates an interesting question and an interesting experimental paradigm. Abstract could be improved by adding a final 'wrap up' sentence that provides a statement of wider significance.

Introduction

General

I understand that space is tight but suggest reworking introduction to broaden interest and reduce amount of time spent previewing results. Intro is very heavily focused on Drosophila work. Adding in some more information right at the start to set the stage would improve the introduction and potentially broaden reader interest. In a first few sentences or small paragraph can the authors reference work outside Drosophila done to explore the synaptic basis of perceptual decision making? If it has been hard to establish that link casually in other systems, that provides a nice transition to introducing the Drosophila olfactory system as an ideal testbed. Suggest then reducing amount of text previewing results and spend some more text explaining what experiments have led to the present study.

Results and Discussion

General

The results and discussion section is focused and generally well written but is heavily centered on Drosophila work. Broadening the discussion just a bit to include more mention of work in other systems, including humans could help to enrich the paper and broaden interest. To be clear, a few additional sentences or short paragraph would be sufficient in my opinion.

Specific

Paragraphs 1,2

I really appreciate the completely open and honest way in which the authors present the successes and failures of their ChR2 dendritic targeting approaches. It was very interesting to hear the account of how some targeting approaches that should theoretically be rock solid just fail completely for no apparent reason. Many other authors would not have the courage to present any result that cannot be explained, but it is very helpful for others to know not to go down those routes, even if the results 'defy simple explanation'. I can imagine that a lot of work went on behind the scenes to plan out then generate those constructs only to reach negative (and downright confusing) results. I am glad the work is being presented in full and commend the researchers on their patience, persistence and willingness to be transparent. I suggest though that the authors clearly indicate in the table in Fig 1 which constructs work and which ones failed (perhaps in an additional column) so that readers can quickly get the take home message.

Description of work in Figure 2 and Figure 2

It is somewhat difficult to picture the experiments being done. Suggest adding a panel here or a supplemental figure that provides a schematic or actual photo of the experimental set up. The set up is

described in the methods at the end of the paper, but it would readers understand the work better if there was something up front in here in the results

Description of work in Figure 3 and Figure 4

The description of results is reasonable, and there is very solid work presented; however, the bottom line conclusion (ie that 'subthreshold voltages are the principle information carriers while the decision forms') is still fundamentally in question because as the authors state, it is impossible to know what the membrane potential of KCs is doing in these types of experiments. Their assumption that the membrane potential change must be lower than that observed in their ephys experiments is reasonable, but unproven. It would strengthen the authors case if they could make a stronger link between their modelling work and experimental work and modelling and experimental work in other publications that would bolster their case in more detail. Future experiments that involve hyperpolarization of membrane potentials (eg with GtACR) could be quite informative and I could imagine other possible cleverly constructed experiments that could be informative, but I understand that these types of experiments are beyond the scope of this short communication. These experiments could be pointed towards however and I think it would actually strengthen the paper if the authors were to propose future experiments in their system (or other systems) that could conceivably strengthen the case for subthreshold integration. This paper would then have value as a beacon that points the for future work in this area.

References

Double check to make sure that 'Drosophila' and all other genus and species names are italicized with the first letter of the genus name being capitalized as is standard in the field. Often citation software overlooks those formatting details.

Reviewer #3 (Remarks to the Author):

The authors use "optical microstimulation" to examine evidence accumulation in a decision-making task in *Drosophila*. The experimental design runs parallel to classic experiments from Shadlen and Newsome on decision-making in monkeys. The main finding is that a specific type of KCs (with interesting temporal integration properties) accumulate evidence that guides a decision, since microstimulation flattens the reaction time curve.

The work is beautifully presented, both figures and text. I had only one major point and then a number of smaller things for the authors to consider.

The authors argue that evidence accumulation is in the subthreshold activity of core KCs. So it's important that the stimulation does not directly drive spikes in KCs. But it seems only $n=5$ KCs were recorded for each of different stimulation conditions (Fig 1f). This sample size is rather small, and a more extensive characterization would increase confidence in the results (increase to $n \geq 10$). Particularly because Gal4 drivers typically show heterogeneity in expression level, so some KCs may be more

sensitive to photostimulation than others. It would also help to have some data about the heterogeneity of expression; if it is extremely homogeneous that would be helpful to know.

Otherwise largely minor considerations:

Fig 2 A number of points:

Panel labeling on the figure seems incorrect. It doesn't seem to match the legend or the main text as far as I can see. Figure columns should be a,b and c,d to match (currently columns are instead a,c & b,d)

The authors should consider removing the brain schematic on top of Fig2. It creates the impression that Chrimson localization is all-or-none when Fig1 shows us it is only a 4-5x change. Also it's not really related to the results presented which are reaction times & choice accuracy. For me it would be more valuable to see a schematic of the behavioral apparatus that conveys what the 'decision zone' looks like and that captures how reaction time is measured. From the text alone it is not clear to me exactly what the decision zone is (where are the borders) and when the 25ms stimulation comes on relative to the position of the fly in the 'decision zone'. Simply the moment it crosses? Also it would help to clarify what 'choose left' means and why it's significant. A schematic could help, and/or more description in the text.

Throughout Fig 2 the no-light datapoints are gray but the fit line is black. It would be better to make them the same color (as done for the pink & gray points & fits).

IMPORTANT: Fig 2c I tried zooming but still wasn't entirely sure the gray datapoints were actually plotted. Are they underneath the other ones? It doesn't look like any of the individual gray points could be significantly different from the pink/red but are the authors claiming they are different?

The text describes a 'marginal at best' effect on choice accuracy, but the legend here says "Bayesian inference offers decisive support for a photostimulation effect on reaction time and choice accuracy (Bayes factor < 0.0001; see Fig.4d)."

It is hard to believe there is a significant effect on choice accuracy for the GluR1 opsin experiments since the gray datapoints seem to always overlap with the pink/red ones. (there's a bit of space between the fit lines, but surely a better metric is whether some actual datapoints are significantly different from one another. If you can't be sure the means of each datapoint are different then surely you can't conclude that the fit can be considered different.)

I am not particularly familiar with Bayes Information Criterion, but are the posterior probability distributions essentially a measure of variance in the quality of model fitting? If so I think that's preferable to show than the single bars in panels b,d,e. When I read those plots, I wanted to know if you have the confidence to accurately measure differences in model fitting and showing distributions is more persuasive to me. Related to this, it wasn't clear if the models are fit on decision sequences from individual flies (in which case you'd have many BIC measures & could put error bars) or all flies are lumped together. I may have missed some description in the Methods, but it would be nice to have a bit more detail in the main text, and particularly to address any variability in the quality of model fitting.

First sentence of The case for subthreshold integration section:

“Restricting CsChrimson to the dendrites...” Restricting is too strong a word, enriching would be more accurate and avoid perpetuating the oversimplification that the localization is all-or-none.

Response to Reviewers of Manuscript NCOMMS-22-49384-T

We thank the reviewers for their comments.

Reviewer 1

Wong et al., investigated evidence accumulation in the Drosophila mushroom body. They have previously shown that a specific subset of Kenyon Cells is required for odor discrimination and consists of ON/OFF cell types that integrate odor increase or decrease. Now, they set out to causally prove the evidence accumulation mechanism in these KCs. Therefore, they made a new optogenetic tool for activating only the dendritic region of the KCs. They showed before that once the cells were spiking, they seemed to have reached the decision threshold. Manipulation of the membrane potential via the new optogenetic tool however only provides subthreshold activation in the KC soma, thus pushing the cell towards spiking but not initiating spiking itself. Flies were tested for odor intensity discrimination with varying difficulty levels. The more difficult the task, the longer the reaction time and the lower the decision accuracy. Interestingly, upon activation of the KC dendrites during the most difficult tasks, the flies reduce their reaction time but still perform with the same accuracy. Activating all compartments of the KCs however, results in a general reduction in reaction time and a lower accuracy. The authors further investigated how optogenetic stimulation affects evidence accumulation by comparing their data to different computational models. Within a drift-diffusion model, the manipulation seems to be best explained by a lowering of the decision threshold.

This manuscript is written very clearly, and the data is presented in an understandable way. The authors fabricated new and clever tools to elegantly manipulate subthreshold activity in their cell of interest. The new data very nicely shows how the accumulation of subthreshold neural activity can be used as an evidence accumulation system and might be a general mechanism across phyla. Thus, I recommend this manuscript for publication after a very minor revision.

Minor comments:

The figure panel description in Figure 2 seems to be mixed up, please check!

Thank you for spotting this mistake, which has been corrected.

Reviewer 2

Overview

This paper sets out to address a very interesting question: how does the process of synaptic integration underly decision making. The model organism is very well suited to this question and the experiments are done carefully and thoughtfully, and then complimented with insightful modelling work. The paper is compact and streamlined but represents a substantial amount of groundwork in genetic tool development that is quite important for the field. Overall, this work could provide an important benchmark and resource for future work in Drosophila and other model organisms.

Abstract

General

Abstract is well written and articulates an interesting question and an interesting

experimental paradigm. Abstract could be improved by adding a final ‘wrap up’ sentence that provides a statement of wider significance.

The abstract now includes a wrap-up sentence.

Introduction

General

I understand that space is tight but suggest reworking introduction to broaden interest and reduce amount of time spent previewing results. Intro is very heavily focused on Drosophila work. Adding in some more information right at the start to set the stage would improve the introduction and potentially broaden reader interest. In a first few sentences or small paragraph can the authors reference work outside Drosophila done to explore the synaptic basis of perceptual decision making? If it has been hard to establish that link casually in other systems, that provides a nice transition to introducing the Drosophila olfactory system as an ideal testbed. Suggest then reducing amount of text previewing results and spend some more text explaining what experiments have led to the present study.

A revised and expanded introduction places our work in a broader context. The preview of the experimental approach has been moved to the beginning of the results section.

Results and Discussion

General

The results and discussion section is focused and generally well written but is heavily centered on Drosophila work. Broadening the discussion just a bit to include more mention of work in other systems, including humans could help to enrich the paper and broaden interest. To be clear, a few additional sentences or short paragraph would be sufficient in my opinion.

We discuss an electrical microstimulation experiment during motion discrimination in the monkey but are unaware of any directly related work in humans.

Specific

Paragraphs 1,2

I really appreciate the completely open and honest way in which the authors present the successes and failures of their ChR2 dendritic targeting approaches. It was very interesting to hear the account of how some targeting approaches that should theoretically be rock solid just fail completely for no apparent reason. Many other authors would not have the courage to present any result that cannot be explained, but it is very helpful for others to know not to go down those routes, even if the results ‘defy simple explanation’. I can imagine that a lot of work went on behind the scenes to plan out then generate those constructs only to reach negative (and downright confusing) results. I am glad the work is being presented in full and commend the researchers on their patience, persistence and willingness to be transparent. I suggest though that the authors clearly indicate in the table in Fig 1 which constructs work and which ones failed (perhaps in an additional column) so that readers can quickly get the take home message.

The construct bearing the effective localization motif derived from GluR1 is now indicated in red type in Fig. 1a.

Description of work in Figure 2 and Figure 2

It is somewhat difficult to picture the experiments being done. Suggest adding a panel here or a supplemental figure that provides a schematic or actual photo of the experimental set up. The set up is described in the methods at the end of the paper, but it would readers understand the work better if there was something up front in here in the results.

The new Supplementary Fig. 2 shows such a schematic.

Description of work in Figure 3 and Figure 4

The description of results is reasonable, and there is very solid work presented; however, the bottom line conclusion (ie that ‘subthreshold voltages are the principle information carriers while the decision forms’) is still fundamentally in question because as the authors state, it is impossible to know what the membrane potential of KCs is doing in these types of experiments. Their assumption that the membrane potential change must be lower than that observed in their ephys experiments is reasonable, but unproven. It would strengthen the authors case if they could make a stronger link between their modelling work and experimental work and modelling and experimental work in other publications that would bolster their case in more detail. Future experiments that involve hyperpolarization of membrane potentials (eg with GtACR) could be quite informative and I could imagine other possible cleverly constructed experiments that could be informative, but I understand that these types of experiments are beyond the scope of this short communication. These experiments could be pointed towards however and I think it would actually strengthen the paper if the authors were to propose future experiments in their system (or other systems) that could conceivably strengthen the case for subthreshold integration. This paper would then have value as a beacon that points the for future work in this area.

The assumption that optogenetically evoked quanta cause only subthreshold KC depolarizations *in vivo* is supported by strong electrophysiological evidence but cannot be proven formally; the creation of a window for the electrode inevitably also changes the optical path, as we make explicit in our manuscript. No conceivable conceptual model could bolster the case for analog processing, while attempts to constrain a biophysically realistic model would again run into the fundamental problem that the size of the light-induced depolarization cannot be measured in the intact animal without disrupting the integrity of its cuticle and brain.

A drift-diffusion model served as the bridge between membrane potential recordings and reaction time measurements in an earlier study (Groschner et al., Cell 2018). This study formulated the hypothesis that $\alpha\beta_c$ KCs accumulate sensory information in subthreshold membrane potential changes and contrasted subthreshold with spike-dependent mechanisms, a topic taken up again in follow-up work (Vrontou et al, Curr. Biol. 2021). We refer readers to these publications.

The goals of the modelling work in the present manuscript are, first, to discriminate between integration and non-integration strategies of decision-making, and second, to delineate the light-sensitive parameter—and, therefore, the putative site—of integration. These models, like any conceptual model, cannot speak to the question of whether subthreshold or suprathreshold quantities are integrated.

Hyperpolarizing the membrane potential with GtACR is also unlikely to afford a distinction between suprathreshold and subthreshold integration, as the channel's large photocurrents will eliminate both synaptic potentials and spikes. In addition, we discovered that the commonly used GtACR transgenes suffer from significant leaky expression in muscle, which precludes their use in our behavioral experiments. We

therefore prefer not to propose these experiments.

References

Double check to make sure that 'Drosophila' and all other genus and species names are italicized with the first letter of the genus name being capitalized as is standard in the field. Often citation software overlooks those formatting details.

We have checked the formatting of all references.

Reviewer 3

The authors use “optical microstimulation” to examine evidence accumulation in a decision-making task in Drosophila. The experimental design runs parallel to classic experiments from Shadlen and Newsome on decision-making in monkeys. The main finding is that a specific type of KCs (with interesting temporal integration properties) accumulate evidence that guides a decision, since microstimulation flattens the reaction time curve.

The work is beautifully presented, both figures and text. I had only one major point and then a number of smaller things for the authors to consider.

The authors argue that evidence accumulation is in the subthreshold activity of core KCs. So it's important that the stimulation does not directly drive spikes in KCs. But it seems only $n=5$ KCs were recorded for each of different stimulation conditions (Fig 1f). This sample size is rather small, and a more extensive characterization would increase confidence in the results (increase to $n \geq 10$). Particularly because Gal4 drivers typically show heterogeneity in expression level, so some KCs may be more sensitive to photostimulation than others. It would also help to have some data about the heterogeneity of expression; if it is extremely homogeneous that would be helpful to know.

The revised Fig. 1f reports data from 10–14 cells per stimulation condition. The amplitude of the light-evoked depolarization serves as a proxy of CsChrimson expression level; direct measurements of protein expression in single KCs are not feasible.

Otherwise largely minor considerations:

Fig 2 A number of points:

Panel labeling on the figure seems incorrect. It doesn't seem to match the legend or the main text as far as I can see. Figure columns should be a,b and c,d to match (currently columns are instead a,c & b,d)

Thank you for spotting this mistake, which has been corrected.

The authors should consider removing the brain schematic on top of Fig2. It creates the impression that Chrimson localization is all-or-none when Fig1 shows us it is only a 4-5x change. Also it's not really related to the results presented which are reaction times & choice accuracy. For me it would be more valuable to see a schematic of the behavioral apparatus that conveys what the 'decision zone' looks like and that captures how reaction time is measured. From the text alone it is not clear to me exactly what the decision zone is (where are the borders) and when the 25ms stimulation comes on relative to the position of the fly in the 'decision zone'. Simply the moment it crosses? Also it would help to clarify what 'choose left' means and why it's significant. A schematic could help, and/or more description in the

text.

The brain schematics have been removed from Fig. 2 and Fig. 3, and the new Supplementary Fig. 2 shows a schematic of the behavioral setup. Plotting decision accuracies as percentages of choosing the left chamber half is by convention (cf. Fig. 5B in Stine et al., eLife 2020).

Throughout Fig 2 the no-light datapoints are gray but the fit line is black. It would be better to make them the same color (as done for the pink & gray points & fits).

The color scheme is consistent in that gray represents black at reduced opacity.

*IMPORTANT: Fig 2c I tried zooming but still wasn't entirely sure the gray datapoints were actually plotted. Are they underneath the other ones? It doesn't look like any of the individual gray points could be significantly different from the pink/red but are the authors claiming they are different? The text describes a 'marginal at best' effect on choice accuracy, but the legend here says "Bayesian inference offers decisive support for a photostimulation effect on reaction time **and choice accuracy** (Bayes factor < 0.0001; see Fig.4d)." It is hard to believe there is a significant effect on choice accuracy for the GluR1 opsin experiments since the gray datapoints seem to always overlap with the pink/red ones. (there's a bit of space between the fit lines, but surely a better metric is whether some actual datapoints are significantly different from one another. If you can't be sure the means of each datapoint are different then surely you can't conclude that the fit can be considered different.)*

In the Bayesian estimation presented here, the model parameters are fit to the full reaction time and choice accuracy distributions rather than their means. This is important, as reaction time distributions in particular tend to be heavily skewed. The great strength of Bayesian inference is that it quantifies the extent to which the complete set of reaction time and choice accuracy data, under all stimulation conditions, favors one model over another. The downside of the Bayesian approach is that it does not allow statistical statements about where the responsible differences lie—in speed or accuracy, or at specific difficulties of discrimination. The mention in the text of a marginal effect on choice accuracy is a verbal description of subtle differences between the gray and red data points, and the corresponding least-squares fits, that are visible in Fig. 2b,d.

I am not particularly familiar with Bayes Information Criterion, but are the posterior probability distributions essentially a measure of variance in the quality of model fitting? If so I think that's preferable to show than the single bars in panels b,d,e. When I read those plots, I wanted to know if you have the confidence to accurately measure differences in model fitting and showing distributions is more persuasive to me. Related to this, it wasn't clear if the models are fit on decision sequences from individual flies (in which case you'd have many BIC measures & could put error bars) or all flies are lumped together. I may have missed some description in the Methods, but it would be nice to have a bit more detail in the main text, and particularly to address any variability in the quality of model fitting.

The BIC is a measure of the quality of model fitting. It is related to the probability that the model is correct. The absolute value of the BIC is difficult to interpret because it is dependent on, for example, the sample size and the noise in the data. However, when comparing the BIC values of two models fit to the *same* data set, the dependencies cancel and we are left with the ratio of probabilities that one or the other model is

correct, given the data (the Bayes factor). We have added an explanation in parenthesis to the main text and a longer, more technical passage to the methods to make this clearer.

All models in our analysis are fit to the same complete set of reaction time and choice accuracy data of a particular genotype, under all optical stimulation conditions. Because the data sets and resulting differences in Bayesian information criterion (ΔBIC) are large, we can distinguish between models with confidence.

The plots in Fig. 4b, 4d, and 4e compare the fits of different models to the experimental data: drift diffusion vs. extrema detection (Fig. 4b); light-sensitive vs. light-insensitive models (Fig. 4d); and models with light-sensitive bound height vs. light-sensitive scaling factor (Fig. 4e). ΔBIC is a single number that quantifies the level of support for a particular model over another—think of it as a loose Bayesian analogue of the P -value in conventional significance testing where in Fig. 4d, for example, the light-insensitive model is the null hypothesis. ΔBIC is computed from the complete set of reaction time and choice accuracy data of a particular genotype, under all optical stimulation conditions, and is not an average of individual flies.

The posterior distributions in Fig. 4f, in contrast, do not relate to model comparison. They show Bayesian estimates of a single model parameter, bound height A , as a function of light intensity. The estimates were obtained by fitting the experimental data for two genotypes to the same drift-diffusion model with light-sensitive A ; the distributions give the probabilities that A assumes a particular value.

*First sentence of **The case for subthreshold integration** section:*

“Restricting CsChrimson to the dendrites...” Restricting is too strong a word, enriching would be more accurate and avoid perpetuating the oversimplification that the localization is all-or-none.

We agree and have changed the term "restricting" to "targeting."

REVIEWERS' COMMENTS

Reviewer #1 (Remarks to the Author):

All my concerns have been addressed in the revisions.

Reviewer #2 (Remarks to the Author):

The Authors have adequately responded to all reviewer comments. I have no further questions or queries.

Reviewer #3 (Remarks to the Author):

All comments have been thoroughly and helpfully addressed.

Response to Reviewers of Manuscript NCOMMS-22-49384A

Reviewer 1

All my concerns have been addressed in the revisions.

Reviewer 2

The Authors have adequately responded to all reviewer comments. I have no further questions or queries.

Reviewer 3

All comments have been thoroughly and helpfully addressed.

We thank the reviewers for their comments.